# Mononuclear Oxidovanadium(IV) Complexes with BIAN Ligands: Synthesis and Catalytic Activity in the Oxidation of Hydrocarbons and Alcohols with Peroxides

Iakov S. Fomenko [1], Marko I. Gongola [2], Lidia S. Shul'pina [3], Nikolay S. Ikonnikov [3], Andrey Yu. Komarovskikh [1], Vladimir A. Nadolinny [1], Yuriy N. Kozlov [4,5], Artem L. Gushchin [1,*] and Georgiy B. Shul'pin [4,5,*]

[1] Nikolaev Institute of Inorganic Chemistry, Siberian Branch of Russian Academy of Sciences, Prosp. Acad. Lavrentieva, dom 3, 630090 Novosibirsk, Russia
[2] Chemistry Depertment, Novosibirsk State University, 1 Pirogova Str., 630090 Novosibirsk, Russia
[3] A.N. Nesmeyanov Institute of Organoelement Compounds, Russian Academy of Sciences, Ulitsa Vavilova, dom 28, 119991 Moscow, Russia
[4] N.N. Semenov Federal Research Center for Chemical Physics, Russian Academy of Sciences, Ulitsa Kosygina 4, 119991 Moscow, Russia
[5] Chair of Chemistry and Physics, Plekhanov Russian University of Economics, Stremyannyi Pereulok, dom 36, 117997 Moscow, Russia
* Correspondence: gbsh@mail.ru or gushchin@niic.nsc.ru (A.L.G.); shulpin@chph.ras.ru (G.B.S.)

**Abstract:** Reactions of VCl$_3$ with 1,2-Bis[(4-methylphenyl)imino]acenaphthene (4-Me-C$_6$H$_4$-bian) or 1,2-Bis[(2-methylphenyl)imino]acenaphthene (2-Me-C$_6$H$_4$-bian) in air lead to the formation of [VOCl$_2$(R-bian)(H$_2$O)] (R = 4-Me-C$_6$H$_4$ (**1**), 2-Me-C$_6$H$_4$ (**2**)). Thes complexes were characterized by IR and EPR spectroscopy as well as elemental analysis. Complexes **1** and **2** have high catalytic activity in the oxidation of hydrocarbons with hydrogen peroxide and alcohols with *tert*-butyl hydroperoxide in acetonitrile at 50 °C. The product yields are up to 40% for cyclohexane. Of particular importance is the addition of 2-pyrazinecarboxylic acid (PCA) as a co-catalyst. Oxidation proceeds mainly with the participation of free hydroxyl radicals, as evidenced by taking into account the regio- and bond-selectivity in the oxidation of n-heptane and methylcyclohexane, as well as the dependence of the reaction rate on the initial concentration of cyclohexane.

**Keywords:** oxidovanadium complex; BIAN ligands; oxygenation; alkanes; 2-pyrazinecarboxylic acid; alkyl hydroperoxides

## 1. Introduction

In recent decades, some metal complexes have been described as catalysts in alkane oxygenation with dioxygen or peroxides [1–4]. In most oxidation reactions with peroxides, the key oxidizing species is the hydroxyl radical. The first such system that performs oxidation with the participation of hydroxyl radicals is the combination of H$_2$O$_2$ with an iron salt. In the case of Fe(II) it is called "Fenton's reagent" [5,6]. Vanadium coordination compounds have attracted increasing interest due to their structural features [7–18]. Shulpin et al. discovered in 1993 a new, very efficient catalytic system using vanadium complexes in the presence of pyrazinecarboxylic acid with hydrogen peroxide as an oxidizing agent [19–22]. Oxidation mechanisms were later proposed for this system [23,24]. Further studies of various vanadium complexes in the oxidative catalysis of alkanes and alcohols also turned out to be very fruitful [25,26]. These complexes showed high yields of alkane oxidation products. In this case, an important factor is the presence of redox-active ligands in the composition of the complexes. Bis(imino)-acenaphthenes (BIANs) belong to the class of α-diimines, which combine 1,4-diazabutadiene and naphthalene fragments [27–29]. Due to this combination, BIANs have strong σ-donor and π-acceptor properties, providing stabilization of

both high and low oxidation states of the metal upon coordination. BIANs form complexes with almost all main group elements [30–34] and transition metals [35–44]. The key feature of BIANs is their pronounced redox activity, and this property is widely exploited by scientists to implement various catalytic transformations [27]. Historically, the first BIAN-based catalysts were Brookhart's catalysts for the polymerization of olefins [45,46]. The various stereoelectronic properties of BIAN ligands, including their oxidation states, allowed for the modulation of catalyst properties, polyethylene branching, and polymer microstructure [47–49]. Much less attention has been paid to the study of other catalytic processes involving metal/BIAN complexes. The most striking examples are reduction processes, hydrogenation [39,50–55], reduction of nitroarenes [56–58], and hydroamination [33,59–61]. Examples of oxidative transformations catalyzed by metal/BIAN complexes are even rarer, possibly due to the electron-withdrawing properties of ligands [62–67]. There are several examples of vanadium-BIAN complexes that have been tested as catalysts in oxidation reactions. In particular, square-pyramidal V(IV) complexes [VO(acac)(R-bian)]Cl efficiently catalyze the epoxidation of terminal and internal olefins with tert-butyl hydroperoxide or hydrogen peroxide [68] and the related complexes [VOCl$_2$(dpp-bian)] or [VOCl$_2$(dpp-mian)(CH$_3$CN)] provide easy CH-oxidation of alkanes with hydrogen peroxide [62,64]. In this work, we synthesized two new oxidovanadium(IV) complexes with redox-active BIAN ligands [VOCl$_2$(R-bian)(H$_2$O)] (R = 4-Me-C$_6$H$_4$ (**1**) and 2-Me-C$_6$H$_4$ (**2**)) and studied their catalytic properties in the oxidation of cyclohexane with hydrogen peroxide in the presence of 2-pyrazinecarboxylic acid (PCA).

## 2. Results and Discussion

### 2.1. Synthesis of Complexes **1** and **2**

For the synthesis of complexes **1** and **2**, an approach was applied that included the use of vanadium trichloride as a starting compound. During the reaction, vanadium(III) was oxidized in air to form the {VO}$^{2+}$ fragment. Previously, we successfully used this approach to obtain a series of oxidovanadium(IV) complexes with redox-active ligands [62,67,69]. Complexes **1** and **2** were synthesized by a similar method (Scheme 1), by refluxing vanadium trichloride with R-bian (R = 4-Me-C$_6$H$_4$-bian or 2-Me-C$_6$H$_4$) in acetonitrile for 10 h. Fine crystalline powders of complexes **1** and **2** were obtained by recrystallization from a mixture of methylene chloride and hexane in 57% to 49% yields, respectively. Complex **1** is more soluble in most organic solvents than complex **2**.

R = 4-Me-C$_6$H$_4$ (**1**),
2-Me-C$_6$H$_4$ (**2**)

**Scheme 1.** Complex synthesis reaction.

Our attempts to obtain single crystals suitable for X-ray structural analysis for both complexes failed. Therefore, indirect methods were used to determine the composition and structure: elemental analysis, and IR, UV-vis, and EPR spectroscopies. Elemental

analysis data are in good agreement with the proposed formula. The IR spectra of these complexes showed broad vibration bands of the OH group from the coordinated $H_2O$ in the range of 3600–3100 cm$^{-1}$. CH vibrations of the methyl group at 3057–2870 cm$^{-1}$ for **1** and 3055–2869 cm$^{-1}$ for **2** were found. Vibration bands of CC and CN groups of the R-bian ligand appeared in the region of 1661–1018 cm$^{-1}$ for **1** and 1662–1045 cm$^{-1}$ for **2**. Very strong bands at 983 cm$^{-1}$ for **1** and 989 cm$^{-1}$ for **2** were assigned to the VO group [70,71]. The vibrations at 890–818 cm$^{-1}$ for **1** and 870–831 cm$^{-1}$ for **2** can be attributed to the linear chain V = O . . . V = O [69].

The electronic absorption spectra of solutions **1** and **2** in acetonitrile revealed strong absorption in the region of 260–410 nm, which can be attributed to charge transfer bands (involving ligand and metal), as well as a low-intensity band at 497 nm for **1** and 489 nm for **2**, which is typical for d-d transitions.

### 2.2. EPR Spectroscopy Studies

The EPR spectra of **1** and **2** in dichloromethane were recorded at 77 K (Figures 1 and 2). In both cases spectra in solution revealed an eight-line isotropic signal characteristic of V$^{IV}$ (d$^1$) complexes. The spectrum of **1** turned out to be a superposition of two spectra with very close parameters, related to different forms. The ratio between these species was 10:1. The simulation analysis for the strong spectrum (spectrum b) showed the following EPR parameters: $g_1 = 1.96$, $g_2 = g_3 = 1.98$, $A_1 = 17.15$ mT, $A_2 = A_3 = 6.1$ mT, and for the weak EPR spectrum (spectrum c): $g_1 = 1.952$, $g_2 = g_3 = 1.982$, $A_1 = 17.45$ mT, $A_2 = A_3 = 7.5$ mT.

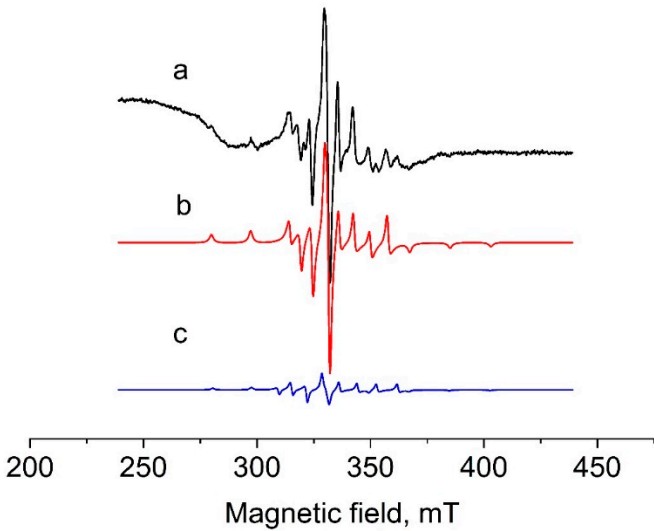

**Figure 1.** EPR spectra of solution of **1** in dichloromethane at 77 K. Black—experimental, red and blue—simulated.

The simulation analysis for the spectrum of **2** gave the following EPR parameters: $g_1 = 1.96$, $g_2 = g_3 = 1.98$, $A_1 = 17.15$ mT, $A_2 = A_3 = 6.1$ mT. These parameters coincided with those for the strong signal in spectrum of **1** (spectrum b), which indicated an identical coordination environment around vanadium. The weak EPR signal with different parameters found in the spectrum of **1** probably belongs to a complex in which vanadium has a different coordination environment. This could be a complex with coordinated acetonitrile [VOCl$_2$(CH$_3$CN)(4-Me-C$_6$H$_4$-bian)], which was formed as a by-product at the stage of synthesis in acetonitrile. In general, the EPR parameters for **1** and **2** are typical for oxidovanadium(IV) complexes [62,72].

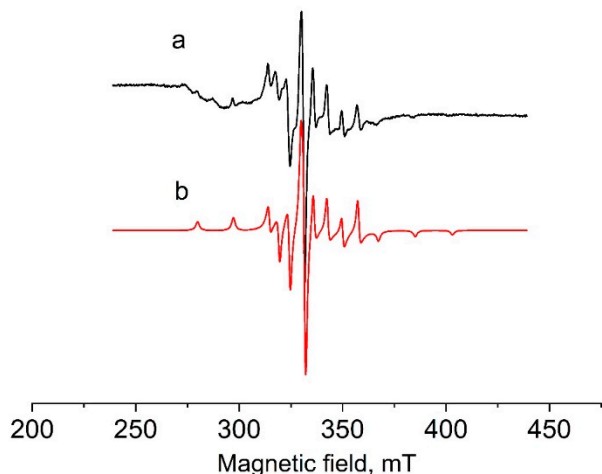

**Figure 2.** EPR spectra of solution of **2** in dichloromethane at 77 K. Black—experimental, red—simulated.

To confirm the composition and structure of the complexes, DFT calculations were carried out. The proposed geometries of the complexes were optimized (Supplementary Tables S1 and S2), and the g-factors and hyperfine interaction (HFI) A tensors were calculated for both complexes.

The calculated EPR parameters were (see Table 1):

**Table 1.** EPR parameters.

| | | | |
|---|---|---|---|
| Complex **1** | g1 = 1.965<br>A1 = 15.8 mT, | g2 = 1.979<br>A2 = 5.4 mT, | g3 = 1.979<br>A3 = 5.4 mT |
| Complex **2** | g1 = 1.965<br>A1 = 15.9 mT, | g2 = 1.979<br>A2 = 5.4 mT, | g3 = 1.981<br>A3 = 5.3 mT |

The calculated values of the EPR parameters are in good agreement with the experimental data, which indicates the legitimacy of the attributed composition $[V^{IV}OCl_2(R\text{-}bian)(H_2O)]$ (R = 4-Me-$C_6H_4$ (**1**), 2-Me-$C_6H_4$ (**2**)). Previously, we obtained a similar complex $[V^{IV}OCl_2(H_2O)(dbbpy)]$ having a 4,4'-di-tert-butyl-2,2'-dipyridine ligand instead of BIAN, for which very similar EPR parameters were found: $g_{xx} = g_{yy} = 1.978$, $g_{zz} = 1.945$, $A_{xx} = A_{yy} = 6.5$ mT, $A_{zz} = 17.86$ mT [69].

### 2.3. Oxidation of Alkanes

We have found that compounds **1** and **2** catalyze the oxidation of alkanes with $H_2O_2$ in acetonitrile in the presence of 2-pyrazinecarboxylic acid (PCA). Accumulation of cyclohexanol and cyclohexanone in oxidation of cyclohexane with hydrogen peroxide catalyzed by compound **1** and **2** is demonstrated by Figures 3–6. The data obtained in the oxidation of cyclohexane for both complexes showed that in the presence of 2-pyrazinecarboxylic acid, the reactions proceeded much faster, which is consistent with our previous studies of oxidative processes using vanadium complexes as the catalysts [25]. A co-catalyst in these reactions was 2-pyrazinecarboxylic acid.

Figure 3 shows the accumulation of products of cyclohexane oxidation with hydrogen peroxide using complex **1** as a catalyst in the absence of 2-pyrazinecarboxylic acid (PCA). Figure 4 shows the accumulation of cyclohexane oxidation products when PCA was added.

Complex **2** was studied in more detail. The reduction of the reaction solution with $PPh_3$ gave rise to a higher concentration of cyclohexanol and a decrease in cyclohexanone concentration (Figure 4) (compare Graphs A and B). These changes indicate (the so-called Shul'pin method [24,62,73]), that alkyl hydroperoxide is formed in the course of the oxidation. Alkyl hydroperoxides are transformed in the GC injector into a mixture of the

corresponding ketone and alcohol. Due to this, we quantitatively reduced the reaction samples with PPh$_3$ to obtain the corresponding alcohol. Shul'pin's method allows us to calculate the real concentrations not only of the hydroperoxide but of the alcohols and ketones present in the solution at a given moment.

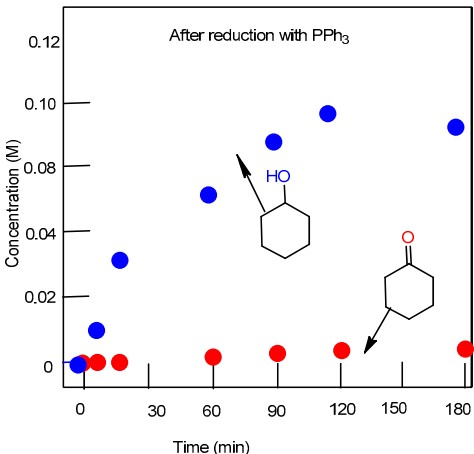

**Figure 3.** Accumulation of cyclohexanol and cyclohexanone in the oxidation of cyclohexane (0.46 M) with H$_2$O$_2$ (2.0 M) catalyzed by complex **1** ($5 \times 10^{-4}$ M) in the absence of PCA, at 50 °C in acetonitrile. Concentrations of products were measured by GC after the reduction of the reaction samples with solid PPh$_3$.

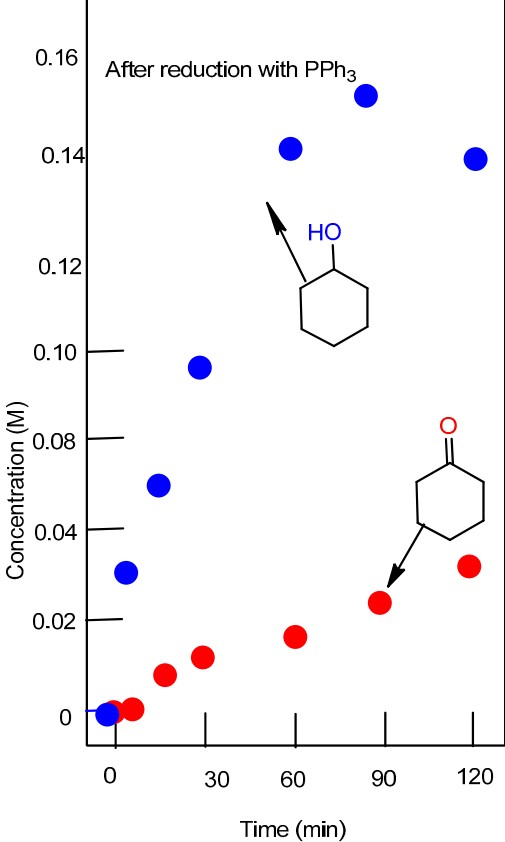

**Figure 4.** Accumulation of cyclohexanol and cyclohexanone in the oxidation of cyclohexane (0.46 M) with H$_2$O$_2$ (2.0 M) catalyzed by complex **1** ($5 \times 10^{-4}$ M) in the presence of PCA, at 50 °C in acetonitrile. Concentrations of products were measured by GC after the reduction of the reaction samples with solid PPh$_3$.

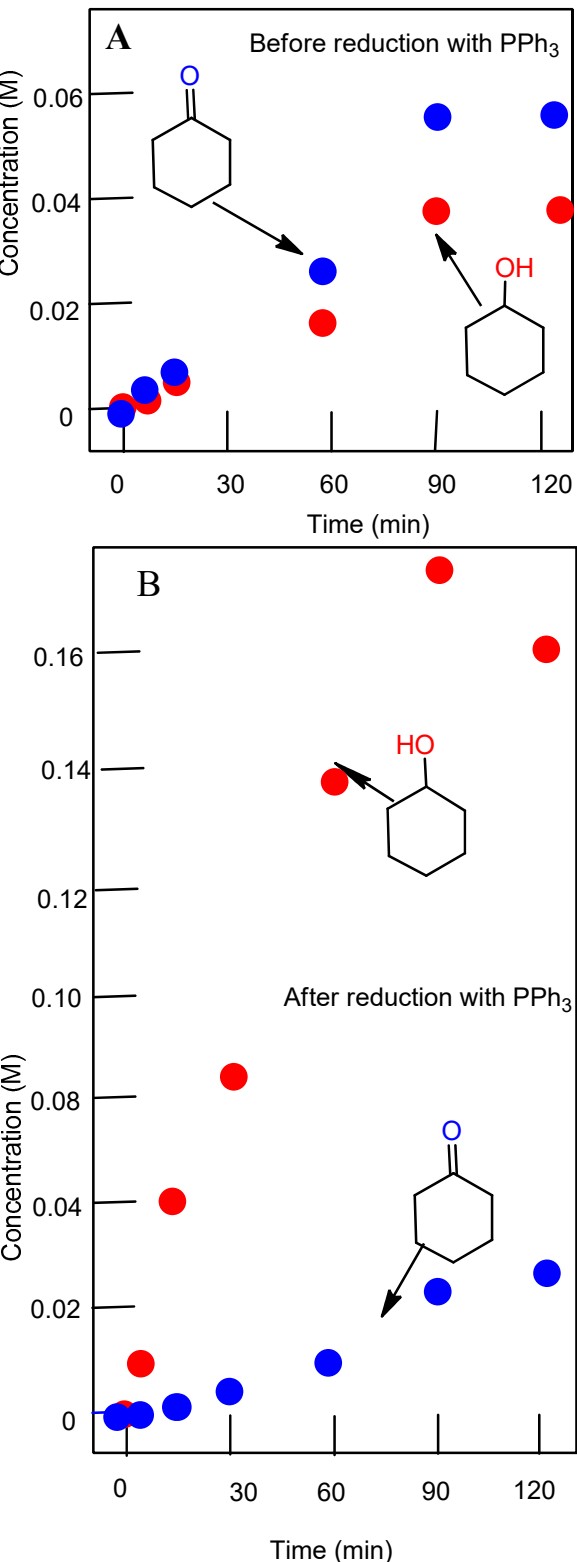

**Figure 5.** Accumulation of cyclohexanol and cyclohexanone in the oxidation of cyclohexane (0.46 M) with $H_2O_2$ (2.0 M) catalyzed by complex **2** ($5 \times 10^{-4}$ M) in the presence of PCA ($2 \times 10^{-3}$ M), at 50 °C in acetonitrile. Concentrations of products were measured by GC before (**A**) and after (**B**) the reduction of the reaction samples with solid PPh$_3$.

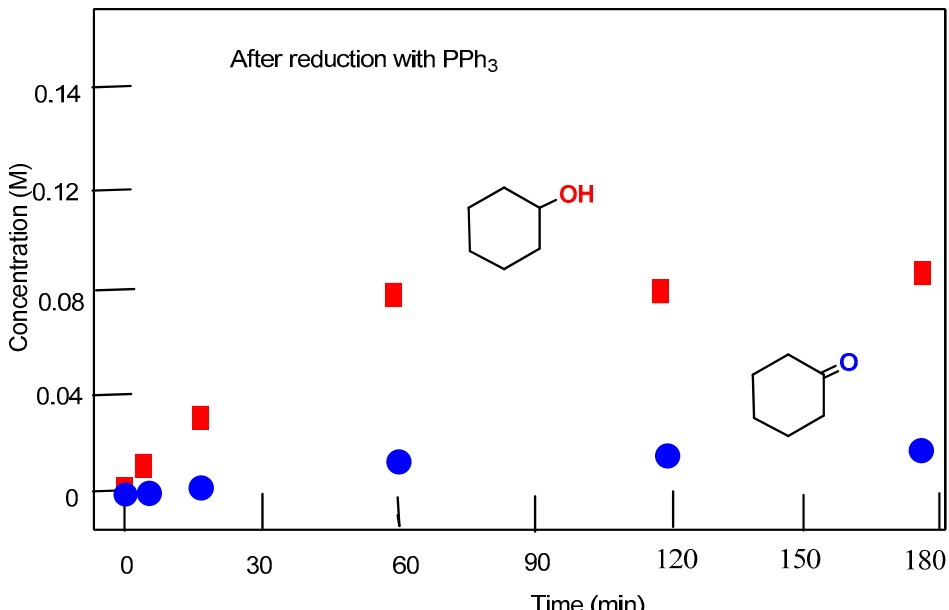

**Figure 6.** Accumulation of cyclohexanol and cyclohexanone in the oxidation of cyclohexane (0.46 M) with $H_2O_2$ (2.0 M) catalyzed by complex **2** ($5 \times 10^{-4}$ M) in the absence of PCA, at 50 °C in acetonitrile. Concentrations of products were measured by GC after the reduction of the reaction samples with solid $PPh_3$.

The curve of dependence of the initial oxidation rate by complex **2** in the case of catalysis approaches a plateau at a cyclohexane concentration of >0.4 M (Figures 7 and 8). The rate at $[CyH]_0$ = 0.1 M is approximately equal to half of the maximum rate.

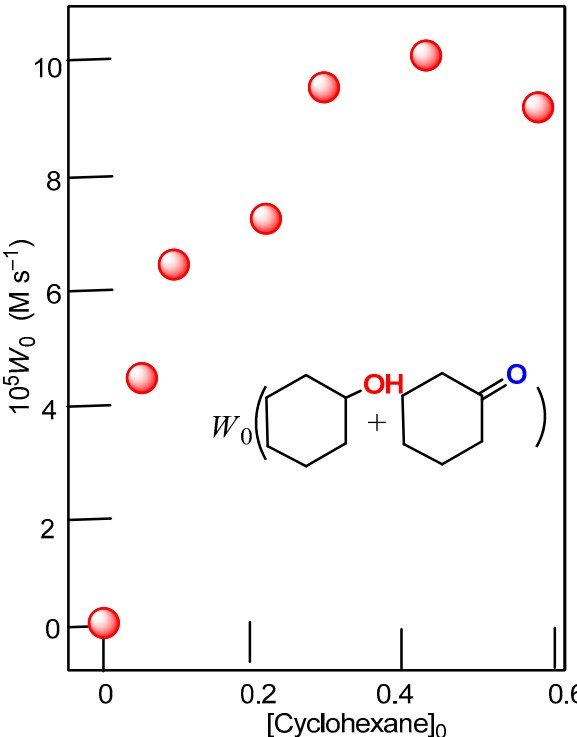

**Figure 7.** Dependence of the initial rate of oxygenate formation $W_0$ on initial concentration of cyclohexane (0.46 M) for the oxidation of cyclohexane with system: complex **2** ($5 \times 10^{-4}$ M)/$H_2O_2$ (2.0 M)/PCA ($2 \times 10^{-3}$ M), at 50 °C in acetonitrile.

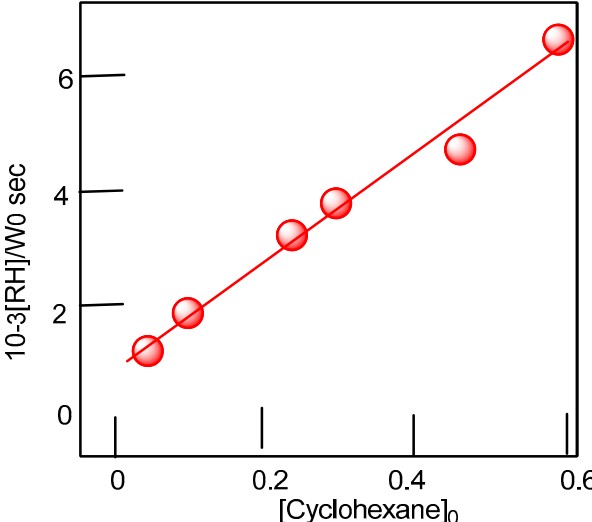

**Figure 8.** Linear anamorphosis of the dependence of the reaction rate on the concentration of cyclohexane (data in Figure 7) in accordance with Equation (11).

We studied the parameters of selectivity in the oxidation of n-heptane (regioselectivity) and methylcyclohexane (bond selectivity) catalyzed by complexes **1** and **2**. The regioselectivity parameters for the oxidation of n-heptane were obtained for complex **1**:C (1):C (2):C (3):C (4) = 1.0:5.6:5.7:5.4 (Complex-**1** ($5 \times 10^{-4}$ M); n-heptane (0.5 M); $H_2O_2$ (2.0 M); PCA ($2 \times 10^{-3}$ M), at 50 °C in acetonitrile (yield of reaction products—16% in 4 h; TON = 160)) and for complex **2**:C (1):C (2):C (3):C (4) = 1.0:5.0:5.2:4.7 (Complex-**2** ($5 \times 10^{-4}$ M); n-heptane (0.5 M); $H_2O_2$ (2.0 M); PCA ($2 \times 10^{-3}$ M), at 50 °C in acetonitrile (yield of reaction products—18% in 3 h; TON = 180)).

The bond-selectivity parameters for the oxidation of methylcyclohexane were obtained: 1°:2°:3° = 1.0:5.5:17.7 for complex **1** (Complex-**1** ($5 \times 10^{-4}$ M); n-heptane (0.5 M); $H_2O_2$ (2.0 M); PCA ($2 \times 10^{-3}$ M), at 50 °C in acetonitrile (yield of reaction products—16% in 4 h; TON = 160)); and 1°:2°:3° = 1.0:6.6:18.5 for complex **2** (Complex-**2** ($5 \times 10^{-4}$ M); methylcyclohexane (0.5 M); $H_2O_2$ (2.0 M); PCA ($2 \times 10^{-3}$ M), at 50 °C in acetonitrile (yield of reaction products—15% in 3 h; TON = 150)).

Data on the selectivity of oxidation (see above) show that the oxidizing species in the studied catalytic system are hydroxyl radicals. This is also confirmed by the data presented below on the reactivity of the oxidizing species. In the system under study, the decomposition of hydroxyl radicals can occur when they interact with the ligand (L) bound to vanadium

$$L + OH^\bullet \rightarrow decomposition \tag{1}$$

with 2-pyrazinecarboxylic acid present in the system

$$PCA + OH^\bullet \rightarrow decomposition \tag{2}$$

or with acetonitrile

$$CH_3CN + OH^\bullet \rightarrow decomposition \tag{3}$$

The interaction of the hydroxyl radical with hydrogen peroxide leads to the formation of the $HO_2^\bullet$ radical, which most likely reduces the vanadium ion

$$H_2O_2 + OH^\bullet \rightarrow H_2O + HO_2^\bullet \tag{4}$$

$$V(5^+) + HO_2^\bullet \rightarrow V(4^+) + H^+ + O_2 \tag{5}$$

The reduced vanadium ion is oxidized by peroxide and the hydroxyl radical is regenerated

$$V(4^+) + H_2O_2 \rightarrow V(5^+) + OH^- + OH^\bullet \tag{6}$$

Thus, the interaction of $OH^\bullet$ with $H_2O_2$ does not in fact lead to the destruction of the hydroxyl radical. Taking the maximum possible value of $10^{10}$ $M^{-1}$ $s^{-1}$ for the rate constants of the interaction of $OH^\bullet$ with L and PCA, we obtained for the rate constants of the pseudo-first order of the decomposition of $OH^\bullet$ with L and PCA under our experimental conditions the values $k_1[L] = 5 \times 10^6$ $s^{-1}$, $k_2[PCA] = 2 \times 10^7$ $s^{-1}$. A similar characteristic for $OH^\bullet$ decomposition with acetonitrile is $k_3[CH_3CN]$ no less than $5 \times 10^7$ $s^{-1}$. Thus, the latter reaction is the main one in the absence of a substrate; the proportion of the reaction involving PCA does not exceed 30% in the total hydroxyl decomposition channel, and the contribution of the reaction involving L is negligibly small. Therefore, the dependence of the initial rate of formation of oxygenation products on the concentration of cyclohexane (Figure 7) reflects the competition of acetonitrile and PCA (reactions (2) and (3)) with the introduced cyclohexane (reaction (7)) for the hydroxyl radical

$$C_6H_{12} + OH^\bullet \rightarrow \tag{7}$$

Let us assume that the rate of generation of hydroxyl radicals by the catalytic system under study under the conditions of Figure 7 is $W_i$. Let us further assume that steps (2), (3), and (7) are limiting the decomposition of $OH^\bullet$ with PCA, acetonitrile, and cyclohexane, and the $OH^\bullet$ concentration is quasi-stationary. Then we can write

$$W_i = (k_2[PCA] + k_3[CH_3CN] + k_7[C_6H_{12}]) \, [OH^\bullet] \tag{8}$$

It follows from (7) that the quasi-stationary $OH^\bullet$ concentration is determined by the relation

$$[OH^\bullet] = W_i / (k_2[PCA] + k_3 \, [CH_3CN] + k_7[C_6H_{12}]) \tag{9}$$

Assuming that reaction (7) is the limiting one in the sequence of transformations leading to the formation of cyclohexyl hydroperoxide, we obtain the following equation for the initial rate of its formation

$$(d[ROOH]/dt)_0 = k_7[C_6H_{12}]_0 W_i / (k_2[PCA]_0 + k_3 \, [CH_3CN]_0 + k_7[C_6H_{12}]_0). \tag{10}$$

To analyze the data in Figure 5, we transform the Equation (9)

$$[C_6H_{12}]_0 / (d[ROOH]/dt)_0 = W_i^{-1} \{ ((k_2[PCA]_0 + k_3 \, [CH_3CN]_0)/k_7) + [C_7H_{12}]_0) \} \tag{11}$$

In accordance with (11), a linear dependence should be observed:
$[C_6H_{12}]_0 / (d[ROOH]/dt)_0$ from $[C_6H_{12}]_0$. The experimental data satisfy the expected linear dependence, from the analysis of which it follows that

$$(k_2[PCA]_0 + k_3[CH_3CN]_0)/k_7 = 0.11 M, \, a \, W_i = 1.1 \times 10^{-4} \, M \, c^{-1} \tag{12}$$

From the above estimates for $k_2[PCA]_0$ and $k_3[CH_3CN]_0$ under the experimental conditions presented in Figure 7, and relation (12), it follows that

$$0.08 < k_3[CH_3CN]_0)/k_7 < 0.11 \tag{13}$$

The estimated value of $k_3[CH_3CN]_0)/k_7$ is close to the values obtained earlier for other catalytic systems in which the formation of hydroxyl radicals has been established [74,75]. Thus, data on the reactivity of an intermediate species of an oxidizing nature that arises during the catalytic decomposition of hydrogen peroxide in the presence of the catalyst under study, as well as data on the regioselectivity of the oxidation of linear alkanes, indicate that the detected intermediate species is a hydroxyl radical.

The dependence of the initial rate of ROOH formation on temperature is shown in Figure 9. It follows from the presented data that the effective activation energy of the process leading to the formation of ROOH equals $18 \pm 2$ kcal/mol. This value is close to the values obtained for the activation energy of alkane oxidation reactions in other previously published works [24,76,77].

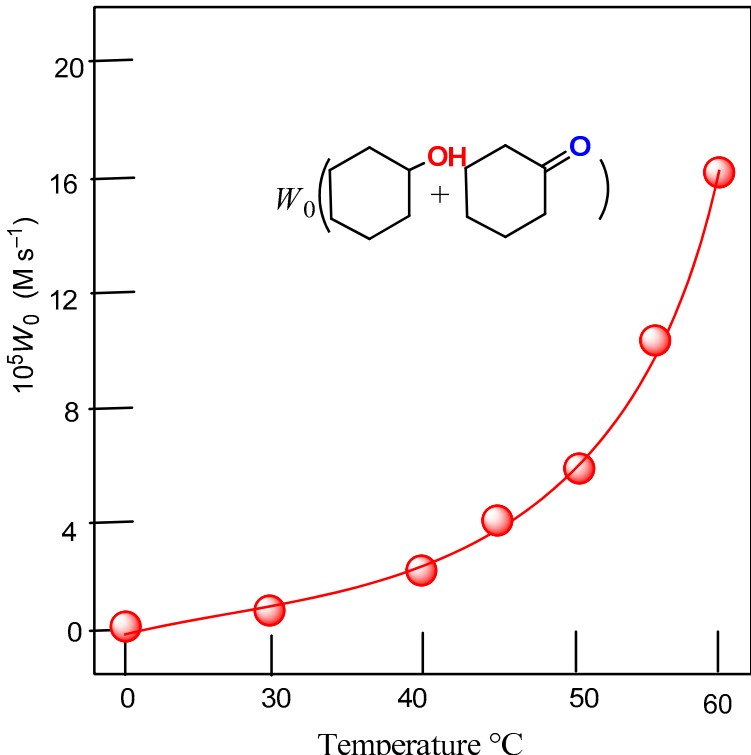

**Figure 9.** Dependence of the initial rate of oxygenate formation $W_0$ on temperature of reaction (conditions of reaction: cyclohexane (0.46 M), complex-**2** ($5 \times 10^{-4}$ M)/$H_2O_2$(2.0 M)/PCA ($2 \times 10^{-3}$ M) in acetonitrile).

These data as well as the character of dependence of the initial cyclohexane oxidation rate on the initial hydrocarbon concentration indicate that the reaction occurs with the participation of hydroxyl radicals, and that alkyl hydroperoxides are formed as the primary products.

Comparative parameters of the oxidation of cyclohexane catalyzed by complexes **1** and **2** and other previously published vanadium complexes are presented in Table 2, and the differences among them are noticeable—the maximum yields of oxidation products are obtained in less time, since the reaction rate is higher, apparently due to the influence of ligands.

**Table 2.** Comparative parameters of the oxidation of cyclohexane catalyzed by complexes 1 and 2 and other previously published vanadium complexes.

| Catalytic System | Total Yield of Oxidation Products (%) | Time (hours) | TON |
|---|---|---|---|
| Complex 1 ($5 \cdot 10^{-4}$ M), this work | 38 | 1.5 | 350 |
| Complex 2 ($5 \cdot 10^{-4}$ M), this work | 42 | 1.5 | 380 |
| $Bu_4N[VO_3]$ ($1 \cdot 10^{-4}$ M) [19] | 30 | 4.0 | 700 |
| $Bu_4N[VO(PCA)_2]$ ($1 \cdot 10^{-4}$ M) [22] | 20 | 24.0 | 900 |
| bis(maltolato)oxo complexes of vanadium(V) ($1 \cdot 10^{-4}$ M) | 27 | 14.0 | 2600 |

**Table 2.** *Cont.*

| Catalytic System | Total Yield of Oxidation Products (%) | Time (hours) | TON |
|---|---|---|---|
| Oxovanadium(V) triethanolaminate ($1 \cdot 10^{-4}$ M) [24] | 10 | 24.0 | 900 |
| bis chelate oxoethoxovanadium ($2 \cdot 10^{-4}$ M) [69] | 30 | 24.0 | 700 |
| [VO(OCH$_3$)(5-Cl-quin)$_2$ 1/2CHCl$_3$ ($5 \cdot 10^{-4}$ M) [68] | 39 | 6.0 | 360 |

### 2.4. Oxidation of Alcohols

The complexes show moderate activity in the oxidation of alcohols with *tert*-butyl hydroperoxide. The yields for the oxidation of phenylethanol (0.5 M) to acetophenone with *tert*-butyl (1.5 M) hydroperoxide under catalysis with complexes **1** ($5 \times 10^{-4}$ M) and **2** ($5 \times 10^{-4}$ M) were 28% (TON = 280) and 56% (TON = 560), respectively, at a temperature of 50 °C, in acetonitrile for 5 h. In analogous reactions of oxidation of cyclohexanol (0.5 M) to cyclohexanone, corresponding yields were 15% (TON = 150) and 20% (TON = 200) after 5 h. In reactions of oxidation of 2-heptanol (0.5 M) to 2-heptanone, corresponding yields were 36% (TON = 360) and 46% (TON = 460) after 5 h. Hydrogen peroxide was much less productive in these reactions.

## 3. Experimental Section

### 3.1. General Procedures

All manipulations were carried out in air. VCl$_3$ was commercially available. 1,2-Bis[(2-methylphenyl)imino]acenaphthene (2-Me-C$_6$H$_4$-bian) and 1,2-Bis[(4-methylphenyl)imino]acenaphthene (4-Me-C$_6$H$_4$-bian) were prepared as reported [78]. Organic solvents (CH$_2$Cl$_2$, MeCN, glacial acetic acid, and hexane) were dried by standard methods before use. All solvents were distilled by standard methods before use.

### 3.2. Physical Measurements

Elemental C, H, and N analyses were performed with a EuroEA3000 Eurovector analyzer. The IR spectra were recorded in the 4000–400 cm$^{-1}$ range with a Perkin–Elmer System 2000 FTIR spectrometer, with samples in KBr pellets and Nujol. EPR spectra were recorded in the X band at 77 and 300 K on an E-109 Varian spectrometer equipped with an analog-to-digital signal converter. To analyze and simulate EPR spectra, EasySpin (Matlab software package) was used [79]. A UV-2501 PC spectrometer was used for UV-vis spectroscopic study. The UV-vis spectra were recorded in a quartz cuvette of 2 mm optical layer at room temperature.

### 3.3. DFT Calculations

The spin-unrestricted DFT calculations were performed using the ADF 2021 program package [80,81]. The optimized geometries were obtained with the generalized gradient approximation (GGA) functional BP86 [82,83] and the triple-zeta basis sets (with one polarization function) TZP [84]. The g- and A-tensors were calculated with hybrid PBE0 functional (25% HF exchange) [85,86] and TZP basis sets for all atoms except V, for which the larger basis set TZ2P-J (triple-zeta with two polarization and several extra tight, mainly 1s, functions) was used [84]. The EPR parameters were derived as second derivative properties with spin-orbit coupling and external magnetic field taken as perturbation [87,88]. In all calculations, the scalar relativistic effects were accounted for by the zeroth-order regular approximation (ZORA) formalism [89,90]. Solvent effects (CH$_2$Cl$_2$) were considered using the conductor-like screening model (COSMO) of solvation [91] as implemented on ADF 2021 [80,81].

Synthesis of [VOCl$_2$(4-Me-C$_6$H$_4$-bian)(H$_2$O)] (1). A mixture of VCl$_3$ (43.6 mg, 277 μmol) and 4-Me-C$_6$H$_4$-bian (100 mg, 277 μmol) was dissolved in 10 mL of acetonitrile. The mix-

ture was refluxed for 10 h. The resulting bright brown solution was evaporated to dryness. Crude product of **1** was redissolved in methylene chloride followed by a layering of hexane. Brown-green crystalline precipitate of **1** formed after 1 day. Yield: 81 mg (57%). Anal. Calc. for $C_{26}H_{22}Cl_2N_2O_2V*H_2O$: C 58.4, H 4.5, N 5.2; Found C 58.6, H 4.8, N 4.9. IR (KBr) $\nu/cm^{-1}$: 3600–3070 (br. s), 3057 (w), 3034 (w), 2970 (w), 2925 (w), 2870 (w), 1964 (w), 1899 (w), 1772 (w), 1726 (w), 1661 (m), 1623 (s), 1586 (vs), 1507 (vs), 1489 (m), 1435 (w), 1419 (m), 1377 (w), 1357 (w), 1312 (w), 1292 (m), 1252 (m), 1226 (w), 1212 (w), 1187 (w), 1150 (w), 1132 (w), 1108 (m), 1064 (w), 1050 (w), 1039 (w), 1018 (w), 983 (vs), 890 (w), 858 (w), 830 (s), 818 (s), 776 (vs), 711 (w), 658 (w), 637 (w), 624 (w), 605 (w), 554 (w), 528 (w), 514 (w), 489 (m), 458 (w), 424 (m). UV-Vis (MeCN): $\lambda(\varepsilon)$ = 271 nm (15254 $M^{-1}$ $cm^{-1}$), 316 (6899 $M^{-1}$ $cm^{-1}$), 410 (2247 $M^{-1}$ $cm^{-1}$), 497 (536 $M^{-1}$ $cm^{-1}$) nm.

Synthesis of $[VOCl_2(2\text{-Me-}C_6H_4\text{-bian})(H_2O)]$ (2). A mixture of $VCl_3$ (43.6 mg, 277 μmol) and 2-Me-$C_6H_4$-bian (100 mg, 277 μmol) was dissolved in 10 mL of acetonitrile. The mixture was refluxed for 10 h. The resulting bright brown solution and brown precipitate were formed. The solution was filtered and evaporated to dryness. Crude product of 2 was redissolved in methylene chloride, filtered again, and followed by a layering of hexane. Brown-green crystalline precipitate of 2 formed after 1 day. Yield: 70 mg (49%). Anal. Calc. for $C_{26}H_{22}Cl_2N_2O_2V*0.2CH_2Cl_2$: C 59.7, H 4.3, N 5.3; Found C 59.4, H 4.5, N 5.4. IR (KBr) $\nu/cm^{-1}$: 3600–3120 (br.s), 3055 (w), 3018 (w), 2976 (w), 2924 (w), 2869 (w), 2579 (w), 1955 (w), 1909 (w), 1843 (w), 1708 (w), 1662 (m), 1619 (vs), 1596 (s), 1586 (vs), 1486 (vs), 1457 (w), 1445 (w), 1435 (m), 1418 (m), 1384 (w), 1357 (w), 1318 (w), 1294 (m), 1243 (m), 1228 (m), 1192 (m), 1155 (w), 1123 (m), 1094 (w), 1045 (m), 989 (vs), 931 (w), 870 (w), 853 (w), 831 (s), 777 (vs), 758 (s), 719 (m), 699 (w), 668 (w), 650 (w), 631 (w), 612 (w), 555 (w), 543 (w), 531 (w), 511 (w), 500 (w), 475 (w), 488 (m), 412 (w). UV-Vis (MeCN): $\lambda(\varepsilon)$ = 268 nm ($\varepsilon$ = 9584 $M^{-1}$ $cm^{-1}$), 316 nm ($\varepsilon$ = 6112 $M^{-1}$ $cm^{-1}$), 404 nm ($\varepsilon$ = 1136 $M^{-1}$ $cm^{-1}$), 489 nm ($\varepsilon$ = 520 $M^{-1}$ $cm^{-1}$) nm.

*3.4. Catalytic Studies*

Total volume of the reaction solution was 5 mL. (**CAUTION**: the combination of air or molecular oxygen and $H_2O_2$ with organic compounds at elevated temperatures may be explosive!) Cylindrical glass vessels with vigorous stirring of the reaction mixture were used for the oxidation of alkanes with hydrogen peroxide or *tert*-butyl hydroperoxide (for alcohols), typically carried out in air in thermostated solution. Initially, a portion of 50% aqueous solution of hydrogen peroxide was added to the solution of the catalyst, co-catalyst (PCA), and substrate in acetonitrile. The aliquots of the reaction solution were analyzed by GC (3700, fused silica capillary column FFAP/OV-101 20/80 $w/w$, 30 m × 0.2 mm × 0.3 μm; argon as a carrier gas. Attribution of peaks was made by comparison with chromatograms of authentic samples). Usually samples were analyzed twice, i.e., before and after the addition portion by portion of the excess of solid $PPh_3$. This method was proposed and used previously by one of us [92,93].

**4. Conclusions**

New oxidovanadium(IV) complexes **1** and **2** were synthesized by reacting vanadium trichloride with BIAN-type ligands (4-Me-$C_6H_4$-bian and 2-Me-$C_6H_4$-bian) in 57% and 49% yields, respectively. These compounds were characterized by elemental analysis and IR and EPR spectroscopy. Compounds **1** and **2** are a powerful catalyst for the efficient oxidation of alkanes with peroxides. Data on the selectivity of oxidation and the nature of the dependence of the initial rate of cyclohexane oxidation on the initial concentration of hydrocarbon, as well as kinetic studies, indicate that the reaction proceeds with the participation of hydroxyl radicals and alkyl hydroperoxides are formed as the primary products.

**Supplementary Materials:** Supplementary materials can be found at https://www.mdpi.com/article/10.3390/catal12101168/s1, Figure S1: UV-spectrum of **1** in the $CH_3CN$, Figure S2: UV-spectrum of **2** in the $CH_3CN$, Table S1: Optimized coordinates for complex **1**, Table S2: Optimized coordinates for complex **2**.

**Author Contributions:** I.S.F.: Investigation, Writing-original draft. Synthesis and description of vanadium complex compounds; M.I.G.: Investigation. Synthesis of ovanadium complex compounds; L.S.S.: Investigation, Writing-original draft. Conducting and describing catalytic experiments on oxidation catalyzed by vanadium compounds; N.S.I.: Investigation. Conducting catalytic experiments; A.Y.K.: Investigation. DFT calculations. Conducting DFT calculations for EPR; V.A.N.: Investigation. Recording and description of EPR spectra; Y.N.K.: Investigation. Proposal and discussion of the mechanism for vanadium catalysed alkane oxidation.; A.L.G.: Conceptualization, Writing-review & editing. Writing and editing the first draft of the article; G.B.S.: Conceptualization, Writing-review & editing. Writing and editing the first draft of the article. Communication with the editor of the journal. All authors have read and agreed to the published version of the manuscript.

**Funding:** Financial support from the Russian Science Foundation (grant No. 22-23-20123) and the government of the Novosibirsk region (contract r-39) is acknowledged. This work was also performed within the framework of the Program for Fundamental Research of the Russian Federation, Reg. No. 122040500068-0.; GC analysis was performed with the financial support from the Ministry of Education and Science of the Russian Federation using the equipment of the Center for Molecular Composition Studies of the A.N. Nesmeyanov Institute of Organoelement Compounds, Russian Academy of Sciences (INEOS RAS). The authors thank the Ministry of Science and Higher Education of the Russian Federation for access to the equipment of the Centre of Collective Usage, NIIC SB RAS (grant No. 121031700315-2).

**Data Availability Statement:** The data presented in this study are available on request from the corresponding author.

**Conflicts of Interest:** The authors declare no competing financial interest.

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
