# Peer review of "Mononuclear Oxidovanadium(IV) Complexes with BIAN Ligands: Synthesis and Catalytic Activity in the Oxidation of Hydrocarbons and Alcohols with Peroxides"

_catalysts, doi:10.3390/catal12101168_

Round 1

Reviewer 1 Report

In this manuscript, the use of BIAN-coordinated vanadyl complexes as catalysts for hydroxide ion generation to be used for oxidation reactivity is described. In general, the work is of reasonable quality and appears to be based on a long-running program in vanadium oxidation chemistry being run by some of the authors. The work is a generally sound technical quality and is appropriate in scope for the MDPI journal Catalysts.

In terms of needed improvements, these are described below as specific points.

Specific points

1. The data given in the plots such as in Figures 4, 5, 6, and 7 include both points as well as what appear to be 'fitted' lines that connect the points. The fitted lines however are not described at all--where do these come from? I wonder if there is strong evidence for any behavior where the amount of product actually decreases toward the end of the measurement time. In other words, should these be plateaus rather than tailing downward? I would consider this issue carefully, and at least describe what the lines mean and where they come from. As this is a mechanistically oriented paper this seems important.

2. The coordinates of the structures calculated by DFT are not included. These should be in a Supporting Information document (I see none) or better in XYZ type files so that other researchers can see the structures that were calculated. These would also be a notable nice substitute in place of XRD results that could not be obtained for the V complexes.

Author Response

We thank reviewer 1 for valuable comments and advice.

Our responses are given below point – by - point.

Reviewer 1 wrote: - The data given in the plots such as in Figures 4, 5, 6, and 7 include both points as well as what appear to be 'fitted' lines that connect the points. The fitted lines however are not described at all--where do these come from? I wonder if there is strong evidence for any behavior where the amount of product actually decreases toward the end of the measurement time. In other words, should

Response of authors:

The points on the curves are taken as statistical average (experimental errors are ±15%), and the curves between them simply connect these points. Curves, of course, can be removed, but such drawings are usually made in articles on catalysis. The drop in cyclohexanol yields is caused by overoxidation. We always observe this phenomenon in many our works on oxidation with hydrogen peroxide, with the formation of diol, carboxylic acid and other products in small quantities. (for example [M. M. Vinogradov, Y. N. Kozlov, D. S. Nesterov, L. S. Shul'pina, A. J. L. Pombeiro, G. B. Shul 'pin, “Oxidation of hydrocarbons with H2O2/O2 catalyzed by osmium complexes containing p-cymene ligands in acetonitrile”, Catal. Sci. Technol., 2014, 4, 3214-3226. DOI: 10.1039/c4cy00492b; A. N. Bilyachenko, M. M. Levitsky , A. I. Yalymov, A. A. Korlyukov, A. V. Vologzhanina, Y. N. Kozlov, L. S. Shul'pina, D. S. Nesterov, A. J. L. Pombeiro, F. Lamaty, X. Bantreil, A. Fetre, D. Liu, J. Martinez, J. Long, J. Larionova, Y. Guari, A. L. Trigub, Y. V. Zubavichus, I. E. Golub, O. A. Filippov, E. S. Shubina, G. B. Shul'pin, “A heterometallic (Fe6Na8) cage-like silsesquioxane: synthesis, structure, spin glass behavior and high catalytic activity”, RSC Adv., 2016, 6, 48165-48180. DOI: 10.1039/C6RA07081G; on: RSC Adv., 2016, 6, 52248. DOI: 10.1039/C6RA07081G.])

Reviewer 1 wrote:  2. The coordinates of the structures calculated by DFT are not included. These should be in a Supporting Information document (I see none) or better in XYZ type files so that other researchers can see the structures that were calculated. These would also be a notable nice substitute in place of XRD results that could not be obtained for the V complexes.

The coordinates of the structures calculated by DFT have been added to the Supporting Information. Two XYZ files have also been created and attached.

Reviewer 2 Report

The authors reported the synthesis of new oxidovanadium(IV) complexes with BIAN ligands as well as their catalytic activity in the oxidation reactions of alkanes and alcohols. The synthesized complexes 1 and 2 exhibited a powerful catalytic activity for oxidation of alkanes tested. This referee considers that the results shown in this study can be of interest for the readers of Catalysts, and that this manuscript should be accepted for publication after addressing the following points.

1) Page 3, lines 82-91

UV-vis spectral measurements of complexes 1 and 2 should be performed to confirm characteristic peaks attributed to BIAN ligands, V(IV) species, and so on. Please add the results in the revised manuscript.

2) Page 9, line 176

Equation (11) is missing in the text. Please confirm and revise the equation number shown in page 10.

3) Page 9, lines 180-183

For the oxidations of n-heptane and methylcyclohexane, total yield of oxidation products, reaction time, and TON should be described in the revised manuscript.

4) Page 11, lines 241-243

The authors should discuss about the difference of the comparative parameters shown in Table 1 to clarify the superiority of complexes 1 and 2 as oxidation catalysts.

5) Pages 11 and 12, lines 244-252

For the oxidation of alcohols, it would be better to also describe TON.

6) Page 12, line 285

Please check the data of elemental analysis. The found value for hydrogen should be within 0.4% of the calculated value.

Author Response

We thank reviewer 2 for valuable comments and advice.

Our responses are given below point – by - point.

Reviewer 2 wrote: 1) Page 3, lines 82-91

UV-vis spectral measurements of complexes 1 and 2 should be performed to confirm characteristic peaks attributed to BIAN ligands, V(IV) species, and so on. Please add the results in the revised manuscript.

Response of authors: UV-vis spectra for complexes 1 and 2 have been added to Supporting Information. A description of the UV-vis spectra has also been added to the manuscript.

Reviewer 2 wrote: 2) Page 9, line 176

Equation (11) is missing in the text. Please confirm and revise the equation number shown in page 10.

Response of authors:

Here we provide an excerpt from the text of the article with Equation 11

 To analyze the data in Fig. 5, we transform the equation (9)

[C6H12]0/ (d[ROOH]/dt)0 =Wi-1{((k2[PCA]0+k3 [CH3CN]0)/k7) +[C7H12]0)}   (11)

Reviewer 2 wrote: 3) Page 9, lines 180-183

For the oxidations of n-heptane and methylcyclohexane, total yield of oxidation products, reaction time, and TON should be described in the revised manuscript.

Response of authors:

We have included the required calculations in the text of the article

Reviewer 2 wrote: 4) Page 11, lines 241-243

The authors should discuss about the difference of the comparative parameters shown in Table 1 to clarify the superiority of complexes 1 and 2 as oxidation catalysts.

Response of authors:

We have included data from other publications to show that our new vanadium complexes are very efficient catalysts in the oxidation of alkane and are on par with other vanadium-containing catalysts. The difference between them is noticeable ‒ the maximum yields of oxidation products are obtained in less time, since the reaction rate is higher, apparently due to the influence of ligands

Reviewer 2 wrote: 5) Pages 11 and 12, lines 244-252

For the oxidation of alcohols, it would be better to also describe TON.

We did it in the text of the article

Reviewer 2 wrote: 6) Page 12, line 285

Please check the data of elemental analysis. The found value for hydrogen should be within 0.4% of the calculated value.

Response of authors: The elemental analysis data were recalculated to the formula C26H22Cl2N2O2V*H2O, in which the dichloromethane solvate molecule was replaced by water molecule. Both can act as solvate molecules, but the formula, taking into account water, better fits the found values.

UV-vis spectral measurements of complexes 1 and 2 should be performed to confirm characteristic peaks attributed to BIAN ligands, V(IV) species, and so on. Please add the results in the revised manuscript.

Response of authors: UV-vis spectra for complexes 1 and 2 have been added to Supporting Information. A description of the UV-vis spectra has also been added to the manuscript.
